# The Role of Supplements and Over-the-Counter Products to Improve Sleep in Children: A Systematic Review

**DOI:** 10.3390/ijms24097821

**Published:** 2023-04-25

**Authors:** Alice Innocenti, Giuliana Lentini, Serena Rapacchietta, Paola Cinnirella, Maurizio Elia, Raffaele Ferri, Oliviero Bruni

**Affiliations:** 1Child Neurology and Psychiatry Unit, Department of Human Neurosciences, Sapienza University, 00185 Rome, Italy; alice.innocenti@uniroma1.it (A.I.); giuliana.lentini@uniroma1.it (G.L.); serena.rapacchietta@uniroma1.it (S.R.); paola.cinnirella@uniroma1.it (P.C.); 2Oasi Research Institute—IRCCS, 94018 Troina, Italy; melia@oasi.en.it (M.E.); rferri@oasi.en.it (R.F.); 3Department of Developmental and Social Psychology, Sapienza University, 00185 Rome, Italy

**Keywords:** pediatric sleep disorder, treatment, tryptophan, iron, antihistamines, theanine, insomnia, awakenings

## Abstract

The sleep–wake cycle is a complex multifactorial process involving several neurotransmitters, including acetylcholine, norepinephrine, serotonin, histamine, dopamine, orexin and GABA, that can be, in turn, regulated by different nutrients involved in their metabolic pathways. Although good sleep quality in children has been proven to be a key factor for optimal cognitive, physical and psychological development, a significant and ever-increasing percentage of the pediatric population suffers from sleep disorders. In children, behavioral interventions along with supplements are recommended as the first line treatment. This systematic review was conducted, according to the PRISMA guidelines, with the purpose of assessing the principal nutrients involved in the pathways of sleep-regulating neurotransmitters in children and adolescents. Our focus was the utilization of over the counter (OTC) products, specifically iron, hydroxytryptophan, theanine and antihistamines in the management of different pediatric sleep disorders with the intention of providing a practical guide for the clinician.

## 1. Introduction

It has been widely demonstrated that sleep is essential for proper development of cognitive functions, particularly in the first years of life. Several studies have concurred that good sleep is positively correlated with better health overall; it is associated with improved memory, attention, learning, mood and, in general, with better well-being, both at the mental and physical levels [1]. Conversely, inappropriate sleep, either in quality or quantity, is correlated with impaired daytime functioning due to sleepiness, attention deficit, memory dysfunction, both in the short and long term [1,2]. Additionally, inadequate sleep correlates with an increased risk of developing hypertension, obesity, diabetes, depression but also accidents and injuries, and in teenagers, it has been proven to be associated with increased risk of self-harm, suicidal thoughts and suicide attempts [1].

Approximately 25–30% of children suffer from some sort of sleep disorder throughout their childhood with a higher prevalence in children affected by neurological, neurodevelopmental or psychiatric conditions [3,4,5,6,7].

The most common sleep disturbances in infants and toddlers are night-time awakenings, with an incidence of 25–50%, bedtime resistance (10–15%) and difficulty in falling asleep (15–30%) [7]. In older children, sleep onset difficulty is the most frequently reported insomnia symptom (15%) together with sleep-related anxiety (11%) [7].

Sleep is a complex process, regulated by many neurotransmitters and neuromodulators that induce neurochemical changes in the brain to regulate sleep–wake states [8].

The main neurotransmitters involved in the sleep–wake cycle modulation include acetylcholine (ACh), monoamines (such as norepinephrine (NE), serotonin (5-hydroxy-tryptamine, 5-HT), histamine (His), dopamine (DA)), orexin and GABA (g-amino-butyric acid) [8]

Nutrients involved in the pathways of such neurotransmitters have been shown to play a role in sleep state regulation. Although several studies have been published on the role of nutrients in the treatment of pediatric sleep disorders, no systematic review has been conducted on this topic. Our aim was to assess the available evidence on over the counter (OTC) treatments of sleep disorders in children in order to implement their use in clinical practice. Therefore, for the purpose of this study, we focused on nutrients and OTC products and on the role of iron, hydroxytryptophan, theanine and antihistamines in the management of pediatric sleep disorders.

## 2. Materials and Methods

The review was conducted according to the PRISMA guidelines (Figure 1).

One electronic database (PubMed) was systematically analyzed. The search terms were, respectively:iron and sleep and infant OR iron and sleep and child OR iron and sleep and adolescent;histamine and sleep and infant OR histamine and sleep and child OR histamine and sleep and adolescent OR antihistamine and sleep and infant OR antihistamine and sleep and child OR antihistamine and sleep and adolescent;TRP and sleep and infant OR TRP and sleep and child OR TRP and sleep and adolescent;theanine and sleep.

Regarding theanine, no age restriction was applied due to the scarce availability of publications on the subject.

The filters applied were articles published after the year 2000, only humans and in English. According to the PRISMA method, we screened the articles by means of keywords, titles and abstracts first and, subsequently, a full text evaluation was done on selected articles to include the most relevant.

Reviews were excluded while retrospective-cohort, cohort, prospective, observational cross-sectional, case-control, prospective and comparative, multi-center cross-sectional and longitudinal studies were included in the present systematic review.

Exclusion criteria were studies that did not have a clinical implication, not conducted on the pediatric population or that did not evaluate sleep parameters and treatment efficacy.

## 3. Results

Figure 1 illustrates the study selection flowchart.

### 3.1. Iron

Iron is an essential nutrient necessary for producing hemoglobin and a co-factor of tyrosine hydroxylase, which converts tyrosine to dopamine and then, to noradrenaline. It is transported in blood bound to transferrin and, through ferritin, it is stored in monocytic-macrophage cells; hence, ferritin is used as an index of iron peripheral storage. Due to the iron involvement in the synthesis of certain neurotransmitters, the hypothesis has been recently been put forward that it is implicated in the pathogenesis of sleep disorders correlated to restless legs syndrome (RLS) and periodic limb movement disorder (PLMD) and, consequently, iron supplementation has been indicted as a possible treatment [9]

Table 1 summarizes the information on the selected articles dealing with iron supplementation.

#### 3.1.1. Iron and Sleep

Peirano et al. [10] investigated the possible effects of iron deficient anemia (IDA) in infancy and subsequent alterations in sleep organization. They enrolled 55 healthy 4-year-old children; 27 with IDA and 28 nonanemic controls, born full term, and with IDA assessed at 6, 12 and 18 months of age. Both control and experimental groups underwent iron therapy; children diagnosed with IDA at 6 months were administered oral ferrous sulphate 15 mg/day for one year, whereas those diagnosed at 12 or 18 months were supplemented with oral iron 30 mg/day for at least 6 months. A higher percentage of REM sleep was noted in the IDA group, due to both an increase in the number of the single REM stages and of their duration. Furthermore, in the experimental group, within the first sleep cycle, there was a tendency for a shorter first REM latency and for a briefer NREM sleep stage 2 and slow-wave sleep (SWS) duration. The study showed that the occurrence of IDA in early infancy is associated with long-term changes in the temporal organization of sleep stages.

Kordas et al. [13] carried out two similar randomized control trials, in Nepal and Zanzibar, to assess the impact of iron supplementation on sleep in infants by means of maternal reports. The study included 877 infants (age 12.5 ± 4.0 months) from Pemba Island, Zanzibar, and 567 children (aged 10.8 ± 4.0 months) from Nepal who were randomized to receive either iron–folic acid or placebo once a day for 12 months. Sleep quality was evaluated via parental reports on napping frequency and duration, nighttime sleep duration and number of night awakenings. Pemban supplemented children showed longer nighttime sleep duration (1 h) vs. placebo. Nepali iron-integrated infants showed a longer sleep duration vs. placebo but only by 0.3 h at night and 0.4 h in total.

Therefore, it is evident that iron levels have a strong impact on sleep organization even from the first months of life and that it can be safely and effectively used in infants with iron deficiency or with a short sleep duration.

#### 3.1.2. Iron and RLS/PLMD

In 2003, Simakajornboon et al. [11] conducted a prospective study on 39 children (20 males, 19 females, mean age of 7.5 ± 3.1 years) with PLMD. Of the 39 children, 28 had serum ferritin concentration <50 μg/L and were administered iron sulphate at 3 mg/kg/day for 3 months. Among the treated group, 76% of patients showed an improvement in periodic leg movements (PLM) index after 3 months of iron supplementation with a corresponding increase in serum ferritin levels.

An improvement of RLS in 22 children following iron supplementation was reported in 2013 by Tilma et al. [15]. All the participants underwent clinical evaluation and blood tests for serum iron and ferritin levels. The authors noted that RLS symptoms appeared precociously in the first year of life with a corresponding decrease in serum ferritin levels. The main symptoms associated with low ferritin were early awakenings. The children with a ferritin level below 50 ng/mL were administered 5.6 mg iron/kg/day oral iron supplementation with improvement in awakenings and RLS symptoms. Iron treatment positively correlated with a ferritin-concentration-dependent clinical effect.

A retrospective case series evaluated the long-lasting effects of oral iron supplementation in patients with RLS, PLMD or both [16]. Out of 105 patients, aged 10.2 ± 5.3 years, 64 were diagnosed with PLMD, 7 with RLS and 35 with RLS and PLMD in comorbidity. Iron determined a significant improvement in PLMS index (at 3–6 months, 1–2 years and >2 years) and adequate ferritin levels, more than 2 years after iron supplementation.

Another retrospective study [14] analyzed safety, tolerability and efficacy of intravenous (IV) iron sucrose supplementation, as a possible alternative treatment, in pediatric RLS and PLMD patients who had failed to respond or had contraindications to oral supplementation. After a single-dose intravenous iron sucrose (3.6 mg/kg), ferritin serum levels increased from a mean baseline value of 15.3 ± 6.3 ng/mL to 45.7 ± 22.4 ng/mL post treatment. Furthermore, sleep quality improved in 75% of children and only minor adverse events took place in 25% of patients, mainly related to gastrointestinal disturbances or IV catheter placement.

Based on the assumption that the comorbidity between NREM sleep parasomnias and RLS/PLMD in children is rather frequent, and that their association worsens the disease’s burden, Gurbani et al. [17] performed a retrospective study to assess whether treatment of RLS/PLMD with oral iron supplementation determined an improvement also in parasomnia symptomatology. Among 226 children with a diagnosis of RLS/PLMD, 30 patients experienced parasomnia episodes and underwent iron treatment. After iron treatment, 50% of the patients reported an improvement of RLS symptoms and 40% a resolution of the parasomnias. Furthermore, 21 out of the 30 participants underwent polysomnographic evaluation showing a reduction in PLM index and PLMS-related arousals.

A group of researchers evaluated the effect of oral iron in 77 two-to-18-year-old children, with RLS, PLMS/D, OSAS or another sleep disorder: 42 patients were classified as responders with an increase in ferritin values of at least 10 micrograms; 35 were non-responders. The increase in ferritin levels correlated with a decrease in PLMS [20]. Similarly, the IV ferric carboxymaltose (FCM) in 39 patients, 29 with RLS and 10 with PLMD, who either did not tolerate or respond to oral iron supplementation determined an increase in ferritin (from 14.6 ± 7.01 μg/L to 112.4 ± 65.86 μg/L) and in serum iron levels, total iron binding capacity and transferrin levels. Only 14.3% of patients reported adverse effects that were described as mild, mainly gastrointestinal disturbances [24]. The same group [21] conducted an additional retrospective study to compare oral ferrous sulphate (FS) and IV FCM efficacy in pediatric restless sleep disorder (RSD): 15 children received 325 mg as tablets or 3 mg/kg/day as solution of oral iron, and 15 received 15 mg/kg (750 mg maximum) of IV FCM as a single infusion. Despite a statistically significant increase in ferritin levels was noted in both groups, it was higher in the intravenous group.

Finally, an interesting recent case report by Al-Shawwa et al. [23] analyzed the effectiveness of iron infusion therapy in a 2-year-old patient with bedtime resistance, difficulty to settle down, restless sleep and recurrent awakenings that impacted the children’s daytime functioning. The patient was diagnosed with an iron deficient anemia and IV iron treatment resulted in an immediate improvement of the associated sleep disorders.

All these studies concur both on the strict correlation between iron deficiency, RLS and PLMD and on the profound and disturbing effects they have on children’s sleep. Hence, it is evident the importance of assessing iron deficient infants for sleep difficulties and conversely, to screen for iron deficiency children who present with an agitated sleep and multiple night awakenings. Due to the significant beneficial effects of iron therapy on these patients’ symptoms and quality of life, iron oral supplementation should be the first line treatment for RLS and PLMD in the pediatric population and, in the more severe cases or intolerant to oral iron, IV iron therapy can be safely and effectively implemented.

#### 3.1.3. Iron and Autism Spectrum Disorders

Autistic children may also benefit of iron administration: sleep disturbances were noted in 77% of children with insufficient dietary iron intake. After 8 weeks of iron supplementation, the ferritin level increased and sleep quality significantly improved [12].

In 2020, Reynolds et al. [18] carried out a controlled clinical trial in children with autism spectrum disorder, insomnia and low ferritin levels. Twenty patients aged 2 to 10 years were randomly assigned to either 3 mg/kg/day of ferrous sulphate supplementation (*n* = 9) or placebo (*n* = 11) for 3 months. The experimental group showed a decrease in sleep onset latency and awakenings that, however, was not correlated with the increase in iron levels.

This evidence confirms that the correlation between iron profile and sleep disturbances is strong also in children with autism spectrum disorder and support iron supplementation in these group of patients when low ferritin/iron levels and sleep disturbances are noted.

#### 3.1.4. Iron and Psychological Symptoms

Psychological status, as well, may be positively affected by iron administration, as reported in 19 children (aged 6–15 years) with serum ferritin levels < 30 ng/mL and psychological symptoms such as decreased energy, fatigue, insomnia and mood alterations treated with 25–100 mg of oral iron for 12 weeks, alongside psychotherapy. At the 12 week follow up after iron supplementation, an improvement in sleep, general health and psychological status was found [22].

#### 3.1.5. Iron in Sleep Disorder in the Angelman Syndrome

About 20% to 80% of patients with Angelman syndrome are estimated to have sleep disturbances, such as shorter sleep, increased sleep onset latency, and abnormal sleep behaviors. In 15 children with Angelman syndrome affected by insomnia and with serum ferritin level <24 mg/L, iron administration determined improvements in sleep quality [19]. Because of the high frequency of sleep disorder in these patients, it is important to assess them for iron deficiency and to supplement them when needed.

### 3.2. Antihistamines

Histamine is a neurotransmitter that promotes wakefulness. The activity of the histaminergic system is maximal during alert wakefulness, is decreased during quiet wakefulness, and completely suppressed during somnolence, NREM and REM sleep. Histaminergic neurons are in the posterior hypothalamus (within the tuberomammillary nucleus) and project primarily to H1 and H3 receptors in the perifornical hypothalamus (orexin-rich) and cholinergic neurons. First-generation H1 antihistamines are the most used agents for pediatric insomnia, as they reduce sleep latency and nocturnal awakenings and have minimal effects on sleep architecture. They include ethanolamines (such as diphenhydramine) and piperazine derivatives (such as hydroxyzine), as well as trimeprazine and niaprazine [25]. The summary of articles dealing with antihistamines is reported in Table 2.

#### 3.2.1. Antihistamine (Diphenhydramine) and Nighttime Awakenings

“The Trial of Infant Response to Diphenhydramine” (TIRED) is a double-blind, randomized, controlled clinical trial to evaluate the efficacy of diphenhydramine hydrochloride therapy (1 mg/kg) in children aged 6 to 15 months with frequent nocturnal awakenings reported by parents. According to an initial parental report, by day 14, there was an improvement in the number of nocturnal awakenings requiring parental assistance, without adverse effects. However, the study was later stopped due to lack of efficacy of diphenhydramine versus placebo [26].

#### 3.2.2. Use of Antihistamines as Medications for Children with Sleep Difficulties

Wesselhoeft and colleagues [28] conducted a study to describe the use of hypnotic drugs (melatonin, z-drugs, and sedating antihistamines) among 5-to24-year-old subjects in Sweden (2,372,337), Norway (1,295,114) and Denmark (1,397,324). Sedating antihistamines (H1-receptor antagonists including alimemazine, promethazine, and promethazine) were used as hypnotic drugs in children and adolescents. The annual prevalence of sedating antihistamine use was highest in Sweden, 13/1000 in 2018; 7.5/1000 in Norway; and 2.5/1000 in Denmark.

#### 3.2.3. Antihistamine (Hydroxyzine) and Bruxism

Hydroxyzine has been reported to be more effective than placebo in 30 children with bruxism [27]. During a 4-week trial, subjects took hydroxyzine 25–50 mg/day or placebo, orally, at bedtime. No serious side effects were reported. Some explanations for the effectiveness of hydroxyzine are the increase in sleep depth, the reduction of anxiety and the induction of muscle relaxation [27].

### 3.3. Tryptophan

Tryptophan (TRP) is an essential amino acid that can only be acquired through diet and is converted into L-5-hydroxytryptophan (5-HTP) by tryptophan hydroxylase (TPH) that is then decarboxylated to serotonin (5-hydroxytryptamine, 5-HT) which is further processed into melatonin (N-acetyl-5-methoxytryptamine). Therefore, it can be deducible how tryptophan may play a role in sleep regulation. Table 3 provides a summary of the articles dealing with tryptophan supplementation.

#### 3.3.1. Tryptophan in Diet and Sleep

Several studies evaluated the effect of tryptophan enriched diet or milk formulas reporting a moderate but still significant improvement in sleep of infants and children.

Aparicio et al. [29] showed that, after administering TRP-enriched milk at night, the infants slept more, had better sleep efficiency, more immobility time and fewer night movements and awakenings. Furthermore, an increase in 5-hydroxyindoleacetic acid (5-HIAA) and 5-HTP urinary levels were also found, suggesting that the improvements in sleep time and quality were correlated to an augmented synthesis of both serotonin and melatonin, in turn correlated with higher TRP levels administered at night-time.

A clinical trial enrolled 30 infants aged between 8 and 16 months with at least three awakenings during the night in whom cereals were administered with a progressive increase in TRP content, with the standard formula milk, reporting an improvement in sleep onset latency and a decrease of the night awakenings [30].

**Table 3 ijms-24-07821-t003:** Summary of articles dealing with tryptophan supplementation.

Study	Design	Objective	Subjects (Age)	Methods	Results
Bruni et al., 2004 [31]	open trial	assess L-5-HTP effects on sleep terrors	45 children (3–10 years)	EEG and sleep diary	at 6 months follow-up 83.9% of children treated with L-5-HTP were sleep terror-free, while 71.4% of children in the comparison group continued to show sleep terrors
Aparicio et al., 2007 [29]	double blind controlled trial	effects of day/night differentiated milk formulas on the sleep-wake cycle	18 infants (12–20 weeks)	TRP-enriched milk. Sleep daily agenda, actigraphy and urine catecholamine and serotonin metabolites	TRP-enriched milk induces an increase in sleep quality and duration, probably due to an increase in serotonin levels
Harada et al., 2007 [32]	cohort study	evaluate the association between morning TRP intake and circadian typology	2279 children (0–15 years)	TRP index, M-E questionnaire	significant positive correlation between TRP index and M-E questionnaire in infants and young elementary school students aged 0–8 yrs. Lower TRP indices correlated with difficulty in both falling asleep and in waking up in the morning, and with tendency to anger and depression
Cubero et al., 2009 [30]	double blind controlled trial	influence of TRP-enriched cereals, adenosine-5′-phosphate, and uridine-5′-phosphate on sleep quality	30 infants (8–16 months)	actigraphy	TRP-enriched cereals improve sleep quality in terms of sleep onset latency and decrease in the awakenings
Nakade et al., 2009 [33]	cohort study	correlation between TRP breakfast intake and sunlight exposure on circadian typology	744 children (0–6 years)	M-E questionnaire	children with nutritionally well-balanced breakfasts tended to be more morning-typed, and woke up and fell asleep at earlier times
Nakade et al., 2012 [34]	cohort study	evaluate the integrated effects of tryptophan and vitamin B6 intake at breakfast and following sunlight exposure on the circadian typology and sleep habits	816 children (2–5 years)	TRP index, vitamin B6 index, M-E questionnaire	positive correlation between M-E index and TRP index only in children exposed to sunlight for longer than 10 min after breakfast
Van zyl et al., 2018 [35]	retrospective	L-TRP as a treatment for non-REM parasomnia	165 children (3–18 years)	PSG and questionnaires	84% of children taking L-TRP experienced improvements in their parasomnia vs. 47% of non-treated

EEG = electroencephalogram; HTP = hydroxytryptophan; M–E = morningness–eveningness; PSG = polysomnography; TRP = tryptophan.

A positive correlation between the index of TRP taken at breakfast (TRP-Index) and the chronotype evaluated with the Morningness–Eveningness (M–E) questionnaire was found in Japanese infants and children aged 0–8 years. Furthermore, the less the breakfast TRP-Indices were, the more frequent it was for children to have difficulty in both falling asleep and in waking up in the morning and the more frequently they manifested anger and depression [32].

Similarly, another study of 744 children (aged 2–6 years) showed that TRP intake at breakfast promoted morning-type circadian typology and better sleep and less anger and depression. Conversely, evening-type children had an increased rate of depression and anger. The effect of protein intake on shifting the circadian typology to morning-type was augmented if the children were exposed to sunlight for 30–60 min in the morning, indicating that sunlight exposure in the morning might accelerate TRP conversion to serotonin during daytime, which in turn could affect melatonin levels at night [33].

In support of the hypothesis that a higher TRP intake at breakfast may promote serotonin synthesis via light stimulation in the morning and have a natural sleep-inducing effect when converted to melatonin at night, Nakade et al. [34] assessed the association of TRP and vitamin B6 intake, and the following exposure to sunlight on the circadian typology and sleep quality in young Japanese children. Both TRP and Vitamin-B6 intake showed a similar correlation with M–E score for children aged 3 to 5 years, but only by children who were exposed to sunlight for longer than 10 min after breakfast.

These results corroborate the evidence from the previous studies of the importance of TRP intake at breakfast for children to be morning-typed, to have higher sleep quality and indirectly, a good health state.

#### 3.3.2. Tryptophan as Sleep Disorder Treatment

Bruni et al. [31] in 2004 were among the first to carry out an open pharmacological trial to verify the efficacy of L-5-hydroxyTRP (L-5-HTP) in 45 children (aged 3.2–10.6 years) with sleep terrors divided in two groups, one taking 2 mg/kg per day of L-5-HTP at bedtime for 20 consecutive days and the control group in the watchful waiting without any treatment. Patients taking L-5-HTP decreased the number of episodes in 93.5% (29 out of 31) vs. 28.6% of the control group; the improvement was maintained after 6 months follow-up.

Similarly, in a retrospective analysis of medical records, children aged 3 to 18 years with primary parasomnia that received L-TRP (daily dose range: 500–4500 mg, mean dose 2400 mg) experienced improvements in their parasomnia symptoms in 84% of cases, compared to only 47% of the non-treated group [35].

### 3.4. Theanine

L-theanine (γ-glutamylethylamide) is a unique non-protein amino acid that can be found in green tea leaves. It has been suggested that, through a mechanism mediated by glutamate receptors, it induces a noteworthy beneficial effect on stress levels and sleep quality in humans. Unfortunately, only few studies have been implemented, and even less evidence on the effect in children is available. Nonetheless, the results published so far seem promising and suggest that the use of L-theanine might be implemented in the clinical practice. The articles dealing with theanine supplementation are reported in Table 4.

#### 3.4.1. Theanine in Children

Only one randomized, double-blind, placebo-controlled trial to evaluate the effect of 200 mg of L-theanine twice a day for six weeks on sleep quality of 98 children (aged 8–12 years), with a diagnosis of attention-deficit/hyperactivity disorder (ADHD), has been published [36]. In the group taking L-theanine, actigraphic sleep and sleep efficiency scores improved with respect to the placebo group. No adverse events were reported, suggesting that a daily dosage of 400 mg of L-theanine can safely and effectively improve sleep quality in children with ADHD.

#### 3.4.2. Theanine in Adults

A double-blind crossover study tested the effects of green tea with lowered caffeine content (LCGT) on stress and quality of sleep of middle-aged individuals. A significant improvement was reported in participants consuming LCGT because of theanine [37]. In parallel, improved sleep quality and reduced stress levels with lowered caffeine content green tea (LCGT) were also demonstrated in the elderly population [38].

Similarly, a crossover and double-blind trial with 200 mg/day L-theanine in 30 healthy adults with stress-related symptoms improved depression, anxiety and sleep onset latency, as well as sleep disturbances, and decreased the use of sleep medication [39].

Different studies were conducted with theanine in association with other compounds; therefore, it is difficult to evaluate the effects of the single compounds on sleep.

Halson et al. in 2020 [40] carried out a double-blind, placebo-controlled crossover experimental study to investigate the effects of an optimized drink containing six ingredients (tart cherry juice, high GI CHO, α-lactalbumin, adenosine-5-monophosphate (5-AMP), valerian and theanine) on sleep quality, reporting a significant reduction in sleep onset latency. Thiagarajah et al. [44], in a randomized, placebo-controlled, crossover and double-blind study, investigated the effects of alpha-s1-casein tryptic hydrolysate containing RLX2™ and L-theanine in 39 adults with poor sleep quality showing improvement in sleep latency, sleep duration (increased by 45 min), habitual sleep efficiency and daytime dysfunction. No adverse events were reported. A combination of TRP, glycine, magnesium, tart cherry powder and L-theanine was administered in a double-blind cross-over trial in 16 participants with decrease of sleep onset latency and increase of total sleep time and sleep efficiency [43]. Furthermore, a combination of magnesium, vitamins B6, B9, B12, rhodiola and green tea/L-theanine administered to 100 chronically stressed otherwise healthy adults showed a good effect on depression and stress, but no differences in sleep quality [42].

A mixture including 500 mg of TRP and 200 mg of L-theanine with L-glutamine, D-phenylalanine, tyrosine, multivitamin/mineral, magnesium citrate, zinc and gamma-linoleic acid improved anxiety, depression and sleep disturbances in a 26-year-old female with post-traumatic stress disorder [41].

Despite the evidence suggesting a potential beneficial effect of L-theanine on sleep, most studies used mixtures of it with several other compounds and therefore, disentangling the specific effect of L-theanine is difficult.

## 4. Discussion

This review highlights that there have been relatively few published reports of randomized placebo-controlled studies evaluating the efficacy and safety of OTC agents in infants and children with sleep disturbances, despite their widespread and frequent use, often without input from a health care provider. Notwithstanding this, our systematic review emphasizes that multiple OTC treatments can be used effectively in children with sleep disorders. Such disorders, as mentioned above, have a high incidence in the pediatric population and a strong impact on daytime functioning. While the first line treatments remain the behavioral interventions, if these fail, or in addition in the more severe cases, OTC agents can be safely prescribed.

Our systematic review of this partial OTC literature should be interpreted in light of specific limitations. We did not include some agents and especially melatonin, as well as other herbal agents, for which there is very limited literature data and that were not commonly used in Italy, such as valerian, passionflower, St John’s wort, kava kava, etc. The exclusion of these agents should be considered when interpreting the evidence summarized in the present study. Finally, we focused on the treatment of sleep disorders or insomnia in otherwise healthy children (although we considered iron in autism based on the literature).

What emerges from this review is that, depending on the prevailing sleep disorder and associated comorbidities, there are indications to use different compounds. Particularly regarding sleep disorders related to RLS, PLMD and in children, IDA, the treatment of choice, supported by convincing evidence, is oral iron supplementation and, in the most severe cases, IV iron supplementation can be used with sufficient safety and effectiveness.

As for children with an anamnesis of allergies, it is known that histamine is a “wake-promoting” agent [45] and increased levels can be the main pathogenic factor of sleep disturbance in these patients, hence the role of antihistamines in treating such disorders. However, we must take into account that tolerance to the sedative effects of antihistamines can develop quickly.

Regarding TRP, there is solid evidence on its significant role in the sleep cycle regulation and efficiency. It represents the first line in the treatment of awakenings and especially of parasomnias, both in terms of effectiveness and safety. Since L-TRP (not L-5-HTP) in 1989 determined eosinophilia myalgia syndrome, 5-HTP has been under vigilance by consumers, industry, academia and government for its safety. However, no definite cases of toxicity have emerged despite the worldwide usage of 5-HTP for several years and there is no evidence of 5-HTP intake as a cause of any illness, especially the eosinophilia myalgia syndrome or its related disorders [46].

Moreover, although there is a scarce literature on theanine in the pediatric population, what emerges from adult studies is that it has a safe and tolerable profile and an effectiveness both in terms of amelioration of quality and quantity of sleep. Nevertheless, further studies are needed to elucidate its mechanism of action and therefore, its clinical applications.

In conclusion, since OTC without a prescription can be used to treat insomnia and other sleep issues in children, there is a need to have clear indications about risk factors, side effects and other potential concerns. It is also important to know that supplements are not regulated by the Food and Drug Administration as strictly as other medications. However, it has been demonstrated that OTC treatments represent, in the majority of pediatric sleep disorders, the safest therapeutic option and their conscientious employment in clinical practice is suggested and sufficiently supported by the literature.

## Figures and Tables

**Figure 1 ijms-24-07821-f001:**
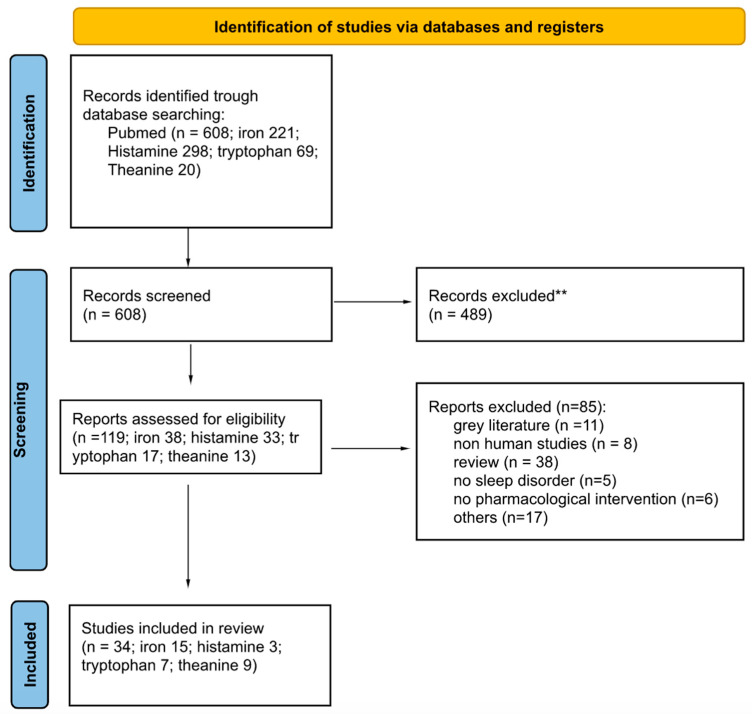
Flow chart of the literature search according to the PRISMA guidelines. ** Records excluded: reviews or studies that did not have a clinical implication, not conducted on the pediatric population or that did not evaluate sleep parameters and treatment efficacy.

**Table 1 ijms-24-07821-t001:** Summary of articles dealing with iron supplementation.

Study	Design	Objective	Subjects (Age)	Methods	Results
Peirano et al., 2007 [10]	retrospective cohort study	association of IDA in infancy with long term alteration of the sleep cycle organization	55 children (4 years)	PSG	IDA in early infancy is associated with long-term changes in the temporal organization of the sleep stages
Simakajornboon et al. 2003 [11]	prospective study	assess relationship between serum iron and ferritin levels and PLMS, and response to supplemental iron therapy	39 children (4–11 years)	PSG, serum iron and ferritin levels	positive correlation between PLMS index and iron levels; 76% of patients improved PLMS index after 3 months of iron supplementation with an increase in ferritin levels
Dosman et al., 2007 [12]	open label	effect of iron supplementation on sleep and serum ferritin levels in children with ASD	33 ASD children (1–8 years)	clinical evaluation, serum ferritin levels, SDSC, PLMS scale	restless sleep score showed improved after iron supplementation
Kordas et al., 2009 [13]	Randomized placebo- controlled trial	effect of iron supplementation on infant’s sleep as reported by the mothers in Nepali and Pemban children	877 Pembans (8–16 months) 567 Nepalis (6–14 months)	maternal reports	longer sleep duration in iron supplemented infants vs. placebo
Grim et al., 2013 [14]	retrospective study	efficacy and safety of IV iron sucrose in children with RLS/PLMD	16 children (2–16 years)	PSG, serum ferritin levels, parent reports	IV iron sucrose can be considered a valid and rather safe alternative to oral iron supplementation
Tilma et al., 2013 [15]	cohort study	define pediatric RLS symptoms and iron supplementation efficacy	22 children (0–8 years)	clinical evaluation, serum iron and ferritin levels, PSG	high PLMS index correlated with serum ferritin levels, iron treatment positively correlated with a ferritin-concentration-dependent clinical effect
Dye et al., 2017 [16]	retrospective study	assessment of long-term effects of iron treatment in pediatric RLS and PLMD	105 children (5–15 years)	iron, ferritin, and PLMS index at baseline and at 3, 6, 12 and 24 months after iron therapy	improvement in PLMS index and ferritin levels >2 years after iron treatment
Gurbani et al., 2019 [17]	retrospective study	impact of iron treatment on parasomnias in children with RLS/PLMD	226 children (3–15 years)	ferritin level and PSG before and after iron treatment	iron therapy correlated with improvement in PLMS index, RLS symptoms and resolution of NREM sleep parasomnias
Reynolds et al., 2019 [18]	randomized placebo- controlled trial	oral ferrous sulphate as treatment for insomnia in children with ASD and low ferritin levels	20 children (2–10 years)	serum iron and ferritin levels, actigraphy and Sleep CGI-S at baseline and after 3 months of iron supplementation	amelioration in Sleep CGI-S
Ryan et al., 2020 [19]	retrospective study	to evaluate the risk of iron deficiency in AS and the efficacy of iron supplementation on correlated sleep disorders	19 AS children (2–10 years)	sleep history, PSG, serum ferritin levels before and after oral, IV or combined iron supplementation	AS patients with increased prevalence of iron deficiency and sleep disturbances (vs. age-matched controls), treatable with iron supplementation
DelRosso et al., 2020 [20]	retrospective study	understanding the causative factors in the treatment response variability of PLMD in children	77 children (2–18 years)	clinical evaluation, PSG, ferritin level	increase in serum ferritin levels in response to oral iron supplementation best predicting factor in evaluating PLMD symptoms alleviation
DelRosso et al., 2021 [21]	retrospective study	appraise efficacy and safety of IV FCM in RLS and PLMD	39 patients (5–15 years)	serum iron and ferritin levels and CGI	IV FCM valid and safe alternative in PLMD non-responders to oral iron supplementation
DelRosso et al., 2021 [21]	retrospective study	compare oral FS and IV FCM efficacy in pediatric RSD	30 children (5–18 years)	serum ferritin, iron profile, video-PSG at baseline and at 3- month follow-up after treatment with either oral FS or IV FCM	IV FCM with greater beneficial effect on pediatric RSD vs. oral FS supplementation
Mikami et al., 2021 [22]	prospective study	determine effects of iron supplementation on psychological status of iron deficient children and adolescents, including sleep difficulties	19 children (6–15 years)	PSQI, CGI-S, POMS and CES-d	after iron treatment, contextually to increase in serum ferritin, significant improvement in PSQI, CGI-S, CES-d scores and in POMS subscales at week 12
Al-shawwa et al., 2022 [23]	case report	effectiveness of iron infusion therapy in an RLS patient with related sleep disturbances	2-year-old child	IV iron supplementation	complete resolution of RLS and sleep-related disorder

AS = Angelman syndrome; ASD = autism spectrum disorder; CES-d = center for epidemiologic studies depression scale; CGI-S = clinical global impression severity; FCM = ferric carboxymaltose; FS = ferrous sulfate; IDA = iron deficiency anemia; IV = intravenous; PLMD = periodic leg movement disorder; PLMS = periodic leg movements during sleep; POMS = profile of mood states 2nd edition youth-short; PSG = polysomnography; PSQI = Pittsburgh sleep quality index; RLS = restless legs syndrome; RSD = restless sleep disorder; SDSC = disturbance scale for children.

**Table 2 ijms-24-07821-t002:** Summary of articles dealing with antihistamine treatment.

Study	Design	Objective	Subjects (Age)	Methods	Results
Merenstein et al., 2006 [26]	double-blind, randomized, controlled clinical trial	to evaluate the efficacy of diphenhydramine hydrochloride therapy in children with frequent nocturnal awakenings	44 infants (6 to 15 months)	parental reports	diphenhydramine no more effective than placebo
Ghanizadeh et al., 2013 [27]	randomized placebo- controlled	to investigate the efficacy of hydroxyzine on sleep bruxism in children vs. placebo	30 children (4–12 years)	VAS and CGI-S at baseline and 4-week post treatment	significant reduction in bruxism severity with hydroxyzine treatment, with respect to placebo
Wesselhoeft et al., 2021 [28]	descriptive study	to investigate the use of melatonin, z-drugs and sedating antihistamines among Scandinavian children and young adults	all Scandinavian children (5–24 years)	public databases from Sweden, Norway, and Denmark	annual prevalence of sedating antihistamine use was highest in Sweden, 13/1000 in 2018; 7.5/1000 in Norway and 2.5/1000 in Denmark. Melatonin the most commonly used hypnotic.

CGI-S = clinical global impression severity; VAS = visual analogue scale.

**Table 4 ijms-24-07821-t004:** Summary of articles dealing with theanine supplementation.

Study	Design	Objective	Subjects (Age)	Methods	Results
Lyon et al., 2011 [36]	randomized, double-blind	L-theanine efficacy on sleep quality of ADHD children	98 children (8–12 years)	actigraphy and PSQI	L-theanine increased sleep time and efficiency in ADHD patients
Unno et al., 2017 [37]	double-blind crossover	low caffeine green tea effect on sleep quality and stress levels of middle-aged individuals	20 adults (44–57 years)	EEG, salivary α-amylase activity	low caffeine green tea reduced stress and improved sleep quality
Unno et al., 2017 [38]	open trial	low caffeine green tea effect on sleep quality in the elderly	10 elderly (85–93 years)	EEG, salivary α-amylase activity	low caffeine green tea improved sleep quality
Hidese et al., 2019 [39]	randomized control trial	effects of L-theanine on stress-related symptoms and cognitive functions in healthy adults	30 adults (36–50 years)	self-rating depression scale, state-trait anxiety inventory-trait, PSQI	significant improvement in sleep onset latency, sleep disturbances and use of sleep medication
Halson et al., 2020 [40]	randomized control trial	validate nutritional intervention on sleep quality	18 adult males (20- 33 years)	PSG, cognitive tests, postural sway, subjective sleep quality questionnaire	nutritional interventions can induce a significant improvement in sleep onset latency
Ross et al., 2020 [41]	case report	nutrients efficacy on mood disorders and sleep disturbances	26-year-old female	self-report	nutritional supplements can ameliorate mood disorders and sleep efficiency
Noah et al., 2022 [42]	randomized control trial	effect of Mg-Teadiola on stress	100 adults (18–65 years)	PSQI	better scores at PSQI on day 56
Langan-Evans et al., 2022 [43]	randomized, double- blind, cross-over trial	effects of a nutritional blend, including L-theanine, on sleep quality	16 adults (21–27 years)	actigraphy, PSQI, consensus sleep diary, KSS	the nutritional blend increased total sleep duration and sleep efficiency
Thiagarajah et al., 2022 [44]	randomized, double- blind, cross-over trial	evaluate effects of alpha-s1-casein tryptic hydrolysate and L-theanine on sleep quality	39 adults	PSQI, heart rate, blood pressure, salivary cortisol, EEG	improvement in PSQI total score, sleep latency, sleep duration, sleep habitual efficiency, daytime dysfunction, and increased total and frontal alpha power significantly

ADHD = attention-deficit/hyperactivity disorder; EEG = electroencephalogram; KSS = Karolinska sleepiness scale; PSG = polysomnography; PSQI = Pittsburgh sleep quality index.

## Data Availability

The data presented in this study are openly available on Pubmed at https://pubmed.ncbi.nlm.nih.gov/.

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
