# Peer review of "The Role of Supplements and Over-the-Counter Products to Improve Sleep in Children: A Systematic Review"

_ijms, 2023, doi:10.3390/ijms24097821_

Round 1

Reviewer 1 Report

The topic of the manuscript is of particular interest because the quality of sleep in infants and young children is essential for proper development. Whereas, sleep is a complex process, regulated by a large number of neurotransmitters that induce neurochemical changes in the brain to regulate sleep and wakefulness. In a short introduction, the authors sufficiently justify the causes and epidemiology of sleep disorders in children, younger and older.

The purpose of this manuscript was to learn about nutrients and OTC products, and in particular the role of iron, hydroxytryptophan, theanine and antihistamines in the treatment of sleep disorders in children.

The authors used a systematic review method according to the PRISMA guidelines.

Figure one shows the correct flow chart of the literature search according to the PRISMA.

There were 34 studies included in the analysis. 

Tables 1-4 summarize information on selected articles on iron supplementation (1), antihistamine treatment (2), tryptophan supplementation (3), and theanine supplementation (4). In addition, under each table, the results of research by other authors are discussed in an abbreviated form in the table. The discussion is brief and summarizes the results of our first systematic review of the efficacy and safety of OTC drugs in infants and children with sleep disorders.

46 bibliographic items were referred to in the text of the work. The authors also listed the limitations of their systematic review.

In my opinion, the manuscript is coherent, understandable and provides new facts in the area of OTC treatment of sleep disorders in infants and children based on scientific evidence.

Author Response

We wish to thank the reviewer for the positive comments.

Reviewer 2 Report

This manuscript discusses the effectiveness and safety of over-the-counter (OTC) agents for sleep disturbances in children, which are widely used but have been evaluated in relatively few randomized placebo-controlled studies. The authors conducted a systematic review of available literature and found that multiple OTC treatments can be effective in treating sleep disorders in children, particularly for restless leg syndrome, periodic limb movement disorder, and iron deficiency anemia. However, there are significant similarities when compared to the following two review papers: Brain Sci. 202010(7), 441 and Sleep medicine reviews 56 (2021): 101406.

 Minor editing of English language required.

Author Response

We thank the reviewer for the comments.

Although there could be some overlaps, the two articles reported are different:

The Brains Science paper is related to children with autism spectrum disorders

The Sleep Medicine Reviews paper  is related to Restless sleep in children: A systematic review 

Respectfully

Reviewer 3 Report

Dear Authors, 

This manuscript is reviewing the utilization of over the counter (OTC) products, specifically iron, hydroxytryptophan, theanine and anti-histamines in the management of different pediatric sleep disorders. The following are my comments and critique: 

General: This study is useful because its main contribution of the paper is pediatricians and psychologists who mainly work on sleep quality in children.

Methods: (1) What are the main reasons for you to focus on iron, hydroxytryptophan, theanine and anti-histamines? I would recommend you to study deeper on the level of vitamin and minerals that related to sleep & relax such as Magnesium. Moreover, (2) the Circadian Rhythms should be considered as the impact of bad sleeping habit/ quality in children. 

Discussion and References: The review paper has cited and discusses using the up-to-dated citation and studies, which is excellent. 

Overall, this is a clear, concise, and well-written manuscript. 

Best regards,

Reviewer

Author Response

We thank the reviewer for the comments.

We focused on iron, hydroxytryptophan, theanine and anti-histamines since they are the most used over the counter products by pediatricians for sleep problems in children

We have already published a paper on the role of vitamin D in sleep problems in children, for the magnesium there are few literature evidence to improve sleep in children since it is often used as add compound in over the counter products to improve sleep (i.e. associated with melatonin or theanine)

We agree that the Circadian Rhythms should be considered as the impact of bad sleeping habit/ quality in children but it is out of the scope of the article since it is better managed with melatonin other than OTC compunds.

Reviewer 4 Report

This is a systematic review conducted with the purpose of assessing the principal nutrients involved in the pathways of sleep-regulating neurotransmitters in children and adolescents.
The article is clear and well organized. However, it can be improved by expanding the discussions, which are too short. I also recommend adding a separate paragraph with the conclusions.
The references are appropriate, the article presents 46 references, being up to date.

Author Response

We thank the reviewer for the positive comments.

Round 2

Reviewer 2 Report

I still have reservations regarding the large overlap. If other reviewers consider it's okay to publish, I respect their opinions.